# Long-Term Monitoring of the Traditional Knowledge of Plant Species Used for Culinary Purposes in the Valencia Region, South-Eastern Spain

**DOI:** 10.3390/plants13060775

**Published:** 2024-03-08

**Authors:** Antonio Belda, Jorge Jordán-Nuñez, Bàrbara Micó-Vicent, Daniel López-Rodríguez

**Affiliations:** 1Departamento de Ciencias de la Tierra y del Medio Ambiente, Universidad de Alicante, Ctra, San Vicente s/n, 03690 San Vicente del Raspeig, Spain; 2Departamento de Ingeniería Gráfica, Universitat Politècnica de València, Plaza Ferrándiz y Carbonell s/n, 03801 Alcoi, Spain; jorjornu@eio.upv.es (J.J.-N.); barmivi@eio.upv.es (B.M.-V.); 3Departamento de Ingeniería Textil y Papelera, Universitat Politècnica de València, Plaza Ferrándiz y Carbonell s/n, 03801 Alcoi, Spain; dalorod@upv.es

**Keywords:** cultural importance index (CI), edible plants, ethnobotany, gastronomy, Mediterranean environments

## Abstract

The aim of this study is to inventory and study ethnobotanical knowledge of edible plants in the Valencian Community (Spain). In respect to culinary uses, 92 species of plant were reported to be edible, finding the following uses: 58 raw, 52 cooked, 16 fried, 7 dried, 21 in liquors and beverages, 25 in dessert and sweets, 11 as seasoning, 17 in pickles, and 10 to curdle milk. We prepared a database that includes genus, family, scientific, and vernacular names in Spanish and Catalan for each plant. We also created a classification of nine edible uses and plant parts used, being Asteraceae (n = 18), Brassicaceae (n = 7), Chenopodiaceae (n = 6), and Rosaceae (n = 6) the families most characterized for gastronomic purposes. The species with the most elevated cultural importance (CI) values were *Foeniculum vulgare* (CI = 1.389), *Cynara scolymus* (CI = 1.374), *Papaver rhoeas* (CI = 1.211), *Beta vulgaris* (CI = 1.167), and *Juglans regia* (CI = 1.155). The most used parts were the leaves (71), flowers (25), and branches (19), while the least used were roots (9) and seeds (8). Traditional knowledge of these plants helps to preserve traditional cuisine, promote the local economy and, in several species, encourage their cultivation.

## 1. Introduction

Throughout the ages, people have interacted with wild and cultivated plants. This relationship has contributed to flourishing of scientific fields such as Geobotany, Economic Botany, Ethnobotany, and Paleoethnobotany [1,2]. In this sense, ethnobotanical research and the interface between researchers and rural populations benefit biodiversity conservation, the expansion of the resident economy, and the recovery of traditional heritage [3]. Therefore, the study and investigation of traditional practices of wild plants and cultured varieties and their derivate products have been progressively increasing over recent decades [4]. In addition, ethnobotanical knowledge includes plants used for treating human and animal diseases and for nutrition, making objects, wood, utensils, construction, hunting, fishing, dyeing, and flavoring plants [5]. Some have symbolic purposes or are used in children’s games [6]. There are toxic plants and species used for smoking, spirits, hedges, and as rootstocks [7,8]. In Spain, specific research has been carried out on edible flora [7,9,10,11,12,13,14], but also some general ethnobotanical research [8,15,16,17]. In the ethnobotanical literature, there are several studies related to the benefits of plants and botany of the Valencia Region [18,19,20]. Thus, during periods of food scarcity like the Civil War and post-war years in Spain, numerous edible wild plants were regularly consumed. Consequently, while many of these plants are not frequently gathered today, some still endure in agroforestry regions [21]. In this sense, due to food scarcity in some places, there was overharvesting of wild plants which led to their diminishing in nature, subsequently declining in their use in recent days [22]. However, the tradition of collecting edible plants continues, and some of them are very common in popular recipes, such as scrambled eggs, soups, rice dishes, salads, pickles or desserts [23]. 

On the other hand, the abandoned and underutilized plant species could become different nutrition resources in the agro-alimentary area, enriching animal and human diets, offering the opportunity for maintainable, resilience to climate change, exploitation, and production with high resistance to pests. However, in the Mediterranean basin, these valuable resources are threatened by climate change, overexploitation of natural resources, and monoculture practices [24]. The benefit of a diet which includes a variety of healthy wild foods is due to the fact that we are descendants of hunter–gatherer societies and have evolved as consumers of wild and cultivated plants. Recently, certain species of wild plants and fungi have received a great deal of attention in Mediterranean countries, and at the same time, the consumption of lesser-known plants that may be significant for prevention of illnesses has been documented. In this respect, there are a large number of edible plants described in Spain with medicinal properties. For example, there are plants for the urinary system (*Asparragus acutifolius*, *Cichorium intybus*, *Daucus carota*, *Foeniculum vulgare* or *Petroselinum crispum*), for skin diseases (*Eryngium campestre* or *Cynara cardunculus*), for the circulatory system (*Crataegus monogyna)*, for colds *(Ficus carica*, *Lavatera arborea* or *Papaver rhoeas*), for stomach problems (*Juglans regia* or *Foeniculum vulgare*) or as stimulant plants (*Allium roseum*) [8]. However, some researchers point out that ingested plants are not necessarily healthy, and some species can even be toxic if ingested commonly or in large quantities [25,26].

Currently, society is in a pivotal moment as the procedure of the ethnobotanical oral transmission process has been interrupted and the most customary heritage is only to be found in the knowledge of elderly persons, and of course this is being gradually lost as such people pass away [27]. Recording ethnological and ecological information proves valuable, yet its collection faces methodological complications, including inadequate research or sampling efforts. The absence of collaboration and knowledge-sharing between disciplines has impeded progress in ethnobotanical research and development [28]. Therefore, only through meticulous design of such investigations can the significance of the data be optimized [29]. The initial hypothesis of this work is that there is a local community knowledge about the traditional gastronomic use of wild and cultivated plant native for the Region of Valencia. Thus, the aim of this study was to compile, describe, and assess data on traditionally used wild edible plants from the Valencian Community. As a secondary objective, we want to show the potential of these plants as “wild foods” which may be helpful to preserve traditional cuisine, reactivate the local economy or in some cases, they could be of interest for novel agronomic and marketable purposes.

## 2. Results

### 2.1. Analysis of Ethnobotanical Data and Ethnological Interviews

#### 2.1.1. Description of the People Interviewed

We interviewed a total of 342 participants whose average age was 65.7 years. Among the individuals interviewed in the Valencian Community, 118 were from the province of Alicante, 110 from Valencia, and 114 from Castellón. Most of the people interviewed have occupations related to the rural environment (farmers, stockbreeders, environmental agents, woodcutters, gardeners, etc.). On the other hand, most of these people have no formal education (55.26%), although some of them do have basic compulsory education (16.08%) and in some cases secondary education (9.38%) or vocational training modules (10.23%). Only a very small percentage corresponds to university teachers (3.21%) or people with an academic degree (5.84%). Moreover, 78.46% of the informants were Catalan speakers, who are also capable of speaking Spanish.

#### 2.1.2. Ethnobotanical Analysis and Results

We compiled an inventory of 92 taxa from 37 botanical families, which are used for different culinary aims. Thus, we present the scientific names of these species, voucher code, the botanical family to which they belong, Spanish and Catalan names, gastronomic uses, plant part used, relative frequency of citation (RFC), cultural importance (CI), cultural value (CV), whether wild or cultivated types were used, and the benefits they bring to people (Table 1). We have registered 105 vernacular names in Spanish and 104 in Catalan. The majority, 79 (85.87%), are wild species, and 13 are cultivated in farmlands and cottages (14.13%), although 24 species can be semi-domesticated (Table 1). 

##### Most Used Families and Genus

There are 37 families represented, being Asteraceae (n = 18), Brassicaceae (n = 7), Che-nopodiaceae (n = 6), and Rosaceae (n = 6) the families most commonly used among the plants employed to different culinary goals (Table 1). 

As for the genus, the most frequently used is *Salvia* (n = 3) and the others that are most relevant are as follows: *Asparagus*, *Cynara*, *Scolymus*, *Sonchus*, *Cistus*, *Plantago,* and *Urtica* (n = 2).

##### Species with Ethnobotanical Importance

The most significant plant species used by participants are *Cynara scolymus*, *Beta vulgaris*, *Ficus carica*, *Cichorium intybus,* and *Asparagus acutifolius*, representing more than 85% of the relative citation frequency (RFC).

Among the species with the greatest cultural importance index, nine plants with values higher than 0.95 for the CI index stand out: *Foeniculum vulgare* (CI = 1.389), *Cynara scolymus* (CI = 1.374), *Papaver rhoeas* (CI = 1.211), *Beta vulgaris* (CI = 1.167), *Juglans regia* (CI = 1.155), *Cichorium intybus* (CI = 1.102), *Ficus carica* (CI = 1.044), *Asparagus acutifolius* (CI = 0.994), and *Salvia rosmarinus* (CI = 0.988). In contrast, the lowest CI are as follows: *Galium setaceum* (CI = 0.108), *Kochia scoparia* (CI = 0.108), *Atractylis humilis* (CI = 0.099), *Suaeda vera* (CI = 0.067), and *Avena sterilis* (CI = 0.038) (Table 1, Figure 1).

On the other hand, the results of importance of use are similar if we use the cultural value (CV) index, the main species being the following: *Foeniculum vulgare* (CV = 0.900), *Cynara scolymus* (CV = 0.732), *Juglans regia* (CV = 0.493), *Beta vulgaris* (CV = 0.472), *Cichorium intybus* (CV = 0.428), *Cynara cardunculus* (CV = 0.412), *Papaver rhoeas* (CV = 0.322), *Ficus carica* (CV = 0.315), and *Asparagus acutifolius* (CV = 0.287) (Table 1).

Most species are collected by informants in the field, but some species can be found in local markets in season. Thus, a total of 15 species have been found that are marketed in the appropriate season: *Petroselinum crispum*, *Cichorium intybus*, *Cynara cardunculus*, *Cynara scolymus*, *Sonchus tenerrimus*, *Borago officinalis*, *Eruca vesicaria*, *Capparis spinosa*, *Beta vulgaris*, *Sedum sediforme*, *Glycyrrhiza glabra*, *Juglans regia*, *Ficus carica*, *Olea europaea,* and *Pinus pinea*.

##### Gastronomic Uses

Firstly, the most common gastronomic use of plants was raw or for salads (group 1) with 58 species. Secondly, the species have been used in cooked dishes (boiled, omelettes, soups, stews, etc.) with 52 species (group 2). The third most important group includes plants used as desserts or sweets, with a total of 25 species (group 6). Other important plants have been used for the production of liquors and beverages with 21 species (group 5). In addition, for salty, 17 plants are pickled with white wine vinegar and marine salt, and used to accompany aperitives (group 8). However, plants used as dehydrated (7), to curdle milk or prepare cheese and related dairy foods (10) or as seasoning (11) are less common (Figure 2). *Foeniculum vulgare* (n = 7), *Cynara cardunculus*, *Cynara scolymus*, *Juglans regia*, *Pinus pinea,* and *Silybum marianum* (n = 5) are the species with a greater variety of uses (Figure 3). 

Below are some of the most popular traditional recipes using wild or cultivated plants in the study area (Figure 4). Thus, the recipes shown have been provided by the people interviewed, but there are different publications on traditional cuisine that include some of these plants [20,30]. These gastronomic proposals include salads, soups, rice dishes, sweets or pickles. One of the most typical dishes is baked rice or “passejat” (RFC = 0.79) which uses the petioles of *Cynara* sp. (popularly called “penques”) mixed with meat, chickpeas, potato, blood sausage, pork fat, and stew stock. In the old days, people used to take this dish to the bakeries or ovens in the villages and that is why it is called “passejat” (walking). Another typical recipe is fig jam (RFC = 0.71), which is made to conserve the excess figs in season and have them for the rest of the year. A soup called “borreta” uses the leaves of *Beta vulgaris* mixed with potatoes, cuttlefish, dried pepper, cod, and egg (RFC = 0.64). A very characteristic dish of the Valencian territory is the “empanadillas” or “pastissets” or “minxos” of wild vegetables that generally use a mixture of *Cichorium intybus*, *Papaver rhoeas*, *Beta vulgaris,* and other edible species (RFC = 0.58). Another culinary proposal is the popular *Sonchus tenerrimus* salad seasoned with olive oil, salt, and vinegar (RFC = 0.52). There is a rice broth using the tender shoots of *Silene vulgaris* mixed with chicken, rabbit, green beans, saffron milk caps, and saffron (RFC = 0.49). Pickled *Sedum sediforme* is a very original and authentic dish that is also traditionally used, marinated in salt, vinegar, and water (RFC = 0.41). Another interesting dish is the wild asparagus soup with shrimp, poplar mushrooms, egg, potato, oregano, and toasted bread (RFC = 0.37). Finally, a typical sweet is walnuts stuffed with sweet moniato paste (*Ipomoea batatas*), which are eaten especially during the Christmas holidays (RFC = 0.35).

##### Ethnobotanical Edible Parts

The parts of the plants used for culinary aims are represented in Figure 5. Thus, the most used part of the plant is the leaves (n = 71), followed by flowers (including its parts, inflorescence, and floral summit) (n = 25), branches (19), fruits (including its parts, fructification, and infructescence) (16) and stems (15), while roots (9) and seeds (8) are less popular.

##### Compare the Reported Used with Internal Level

Using the Jaccard similarity coefficient (Cj), we find the following results between the three provinces of the study area: Alicante and Valencia (Cj = 0.98), Alicante and Castellón (Cj = 0.96), and Valencia and Castellón (Cj = 0.91) (Figure 6).

##### ANOVA Statistical Analysis

There are statistically significant differences between the plant family and the CI importance coefficient. Thus, the *p*-value with a value below the significance level of 0.05, indicates significant differences, although graphically it can be seen that the intervals do not overlap each other in many comparisons. In this sense, the most important families are Papaveraceae, Junglandaceae, Moraceae, and Amaryllidaceae. The rest of the intervals already overlap and have a very similar average importance coefficient (Figure 7).

Depending on gastronomic use, there are also differences when determining the order of importance. The *p*-value is zero in this case indicating a very high dependence. The combinations of uses that most increase the importance coefficient in the calculation are as follows: 1,2,4,5,6,7,8; 1,2,3,6; 1,2,3,8; 1,3,4,5,6 and 1,2,3,5 (Figure 8).

Depending on the part of the plant used, there are also significant differences, with a *p*-value equal to zero. In this case, the prominent uses are as follows: L,B,S,R; L,F,Fr; L,St,F; R,L,F; R,St and R alone (Figure 9).

Finally, the ANOVA shows that there are also significant differences according to the classification by type. The *p*-value is 0.054 and cultivated, semi-domesticated (C, Sd), followed by the cultivated type (C), are the ones that obtain the highest IC value, while wild (W) and wild and semi-domesticated (W, Sd) have a tendency to lower CI (Figure 10).

## 3. Discussion

### 3.1. Importance and Health Benefits of Edible Wild and Cultivated Plants

It has been shown that currently, collection of wild edible plants (WEPs) is a common practice among part of the indigenous inhabitants of rural areas in the Valencia Region. Thus, these WEPs are mostly used by rural persons. Nevertheless, in recent years, the desire for a more natural diet has increased the sales of WEPs in local marketplaces. At the same time, trial gardens for the farming and manufacture of wild plants have been recognized on a regional basis in order to fill the gap in the supermarket [12,31]. Based on the characteristics of both wild and cultivated plants, anti-inflammatory and antioxidant activities are the most commonly observed for these species. This is attributed to the remarkable phenolic compounds present in their preparations, showcasing their significant bioactive properties. Moreover, edible plants can be considered a bioactive food because of their nutritional value and organoleptic properties. They also contribute to alleviating some common diseases. Furthermore, they contain valuable bioactive compounds and they have feasible application in high-quality cuisine [32,33]. In this sense, many of the plants cited in this study have biocomponents implicated in human health according to other research. For example, *Foeniculum vulgare* oils (trans-anethole, fenchone, and limonene) exerted inhibitory effects on the bacterial growth (*Staphylococcus aureus*, *Escherichia coli*, *Pseudomonas aeruginosa*, *Staphylococcus epidermidis,* and *Candida albicans*) and anticancer therapies against certain types of cancer, for example, against breast cancer [34]. In addition, blackberries (*Rubus ulmifolius*) are one of the richest fruit sources of anthocyanins, a class of bioactive phenolic compounds that have been shown to have insulin-sensitizing effects and improve glucose utilization [32]. Also, *Cynara cardunculus* is a plant rich in polyphenols, flavonoids, anthocyanins, phenolic compounds, inulin, coumarins, terpenes, dietary fiber, enzymes, polysaccharides, minerals and vitamins, and it has a wide range of medicinal properties such as antimicrobial (inhibiting the synthesis of nucleic acids in Gram-positive and Gram-negative bacteria), anticancer (human hepatoma cells), antioxidant, anti-inflammatory, hypocholesterolaemic, anti-HIV, cardioprotective, hepatoprotective, and lipid-lowering action [35]. Additionally, *Papaver rhoeas* is used as a sleep aid, laxative, chest pain reliever, throat inflammation reliever, antipyretic, fever reducer, antioxidant, antimicrobial (*Staphylococcus aureus*, *Escherichia coli*, *Klebsiella pneumonia,* and *Candida albicans*) and hepatoprotective. There are anthocyanins, amino acids, carbohydrates, fatty acids, vitamins, phenolic compounds, essential oils, flavonoids, alkaloids, coumarins, organic acids and other compounds, in various parts of the plant [36]. Furthermore, *Juglans regia* has antimicrobial (*Staphylococcus aureus and Klebsiella pneumonia)*, antioxidant, anti-inflammatory, anti-diabetic and it has anticancer activity proven against colon adenocarcinoma, breast adenocarcinoma, glioblastoma, astrocytoma, and melanoma cell lines. It is due to the fact that it contains several secondary metabolites, such as polyphenols, flavonoids, and glycosides [37,38]. On the other hand, *Rumex pulcher* has a folic acid level above 100% of the recommended daily intake. In addition, *Urtica dioica*, *Capsella bursa-pastoris,* and *Eruca vesicaria* contain vitamin C levels above the recommended daily intakes. Moreover, *Celtis australis* and *Urtica dioica* are sources of calcium higher than the 50% recommended daily intake. Finally, *Celtis australis* has magnesium values higher than the recommended daily intakes [14]. The use of WEPs is an intangible part of our heritage that we must protect and conserve. Moreover, its consumption contributes to a varied diet rich in fiber, protein, vitamins, and other beneficial compounds. In addition, they are a natural resource that may help to stimulate, always in a sustainable way, the revitalization of the local economy and ecotourism. This is because the utilization of these plants enables the exploitation of another resource in the local economy, generating economic income and jobs. However, WEPs lack recognition as important contributors to human nutrition in developed countries. In this sense, national and international figures dealing with food and agriculture should increase scientific research and focus on plant diversity, traditional ethnobotanical knowledge, and agricultural and ecological research and foster bioconservation policies and maintainable food production. Furthermore, biological and chemical information is of fundamental importance to assess health profits and physiological properties in order to develop medical surveys concerning their effects on the body, safety, and efficacy [39,40].

Consequently, nowadays the interest of utilization of WEPs as potential sources of cosmeceuticals, pharmaceuticals, nutraceuticals, and probiotics for human well-being is continuously increasing [41].

### 3.2. Comparison with Other Studies

The review of the scientific literature confirms that there are such ethnobotanical studies in nearly every country in Europe and many countries outside of it [42,43]. In respect to plant species of interest, in a large-scale study of a geographically adjacent region, the 10 species (*Foeniculum vulgare*, *Rubus ulmifolius*, *Scolymus hispanicus*, *Silene vulgaris*, *Asparagus acutifolius*, *Chamaerops humilis*, *Portulaca oleracea*, *Mentha* sp., *Salvia rosmarinus,* and *Silybum marianum*) with the highest reported uses coinciding with species analyzed in the present study, are typical of Mediterranean environments and form part of the ethnobotanical cultural legacy of the region [12]. Moreover, in similar studies, raw or cooked vegetable parts are the most frequently used, whereas their use in liqueurs, curds or spices are less common [26,44]. Thus, in other Spanish studies, the results are similar, such as *Rubus ulmifolius* for nutritional uses. However, for *Foeniculum vulgare*, *Ficus carica*, *Malva sylvestris*, *Urtica dioica*, *Crataegus monogyna*, *Fragaria vesca*, *Juglans regia*, *Taraxacum officinale*, *Quercus ilex,* and *Rorippa nasturtium-aquaticum*, our cultural index is much higher than if we compare it with other investigations carried out in the Iberian Peninsula [44,45]. Thus, the use of quantitative and qualitative ethnobotanical indices is relevant for the development of ethnobotany-related studies [46]. It draws attention to *Urtica urens* that is consumed in traditional spring plates in Italy, Belarus, Greece, and other Mediterranean countries, while, in Bulgaria, it is not common as a food plant, and only *Urtica dioica* is employed on a large scale, and the cultivated one has been sold in local markets recently [41]. In addition, another species widely consumed in Europe is *Portulaca oleraceae*, which are grown in home gardens but are considered to have a soft and less crunchy texture [47]. 

If we take into account the most represented families, the results are similar to those obtained in another Mediterranean region located to the north (Catalonia), being Asteraceae (9%), Poaceae (8%), Rosaceae (7%), Lamiaceae (5%), Brassicaceae (5%), Fabaceae (5%), and Apiaceae (5%) [48]. On the other hand, if we compare the botanical families with other studies in European countries, the Amaranthaceae (15.39%) family is also important along with the Asteraceae (15.39%) and Brassicaceae (13.84%) [41]. This can be explained by the fact that these families are the most abundant in cultivated fields and adjacent areas, where informants have easy access to these plants [49].

In our study, most of the edible plants are wild, but if we compare it with another study carried out in peri-urban and anthropized areas, the most used plants are cultivated. The results are similar if we compare the most used plant parts: the fruit and infructescence (36%), followed by leaves (21%), aerial part (15%), and flowers and inflorescences (7%) [48]. In addition, in the Mediterranean region, there is a clear predominance of the use of the aerial portions and a minority of the subterranean parts. This is due to its expensive collection and a lower palatability of subterranean parts in respect to other parts [14,47]. Regarding the mode of consumption, the results are analogous, being the most common way fresh and cooked [48], although there are other studies where seasonings are also important [14,41]. 

If we consider the ingesting of wild and cultivated edible plants, the similarity values for the three Valencian provinces are very high if we compare them with other Spanish regions, where the Cj values range between 0.20 and 0.60 [50].

In this work, we contemplate only plant species cited by the volunteers in the Valencia Region. However, there are more plants with edible benefits in the study area [15,51,52]. 

This study opens the door to prospect investigations, in which this database can be reviewed and extended. This permits recovery of forgotten culinary purposes, highlighting the application of different plants for cooking, flavoring, curdling milk, preparation of liqueurs or desserts, resulting in a very novel contribution to Mediterranean ethnological bibliography. Immaterial heritage of these valuable species may be extremely useful to rejuvenate the local economy and, in several cases, encourage their cultivation. Thus, some potential wild plant species to be cultivated in the study territory are as follows: *Mesembryanthemum crystallinum*, *Salicornia ramosissima*, *Crithmum maritimum*, *Cichorium intybus*, *Sonchus tenerrimus*, *Taraxacum vulgare,* and *Borago officinalis*. The ethnobotanical data provided through personal consultations are appreciated because they allow for a very comprehensive knowledge of natural foods in the study area. This useful information is designed to be used as a guide for other ethnobotanical researchers when studying edible flora in their areas.

### 3.3. The Effects of Gender and Socioeconomic Changes

In our study, the majority of informants are women and they have high ethnobotanical knowledge at a gastronomic level. Thus, women account for 59.36% of those interviewed, having a great historical contribution to traditional cuisine, compared to the typical role of men in rural tasks and their low participation in household chores. The results coincide with other studies where women’s medicinal and food provisioning role and related socio-ecological knowledge are still widespread in the Mediterranean basin [53]. Therefore, in other investigations, similar gendered ethnobiology patterns are detected, concluding that gendered knowledge is highly associated with the division of tasks and responsibilities of people [54]. In the study area, plant collection has been a task traditionally linked to women, but nevertheless in other places of the Southern Spain, wild plant gathering is not a gendered activity [13]. 

More than two decades have passed since the beginning of the study and socioeconomic conditions and customs have changed substantially and many informants have died. In this sense, the so-called acculturation process takes place in the Mediterranean basin, especially in modern urbanized societies, particularly, the acceptance of modern philosophy to the detriment of the traditional one is the highest origin of the loss of this information, which must be available for future generations [55]. However, WEPs can help to rise the modification of food manufacture, adding new plants to our diets with valuable benefits. Thus, local plants should be introduced in the agronomic sector, the development of WEPs cultivation techniques is essential, improved cultivars should be designed for in the future, and the expansion of new gastronomic products that are able to fascinate stakeholders and extant distribution networks are necessary [24]. Despite their healthy properties, consumption of WEPs was neglected during the last years due to modern lifestyles and the shift to other western diets and fast foods. Nevertheless, the surging scientific knowledge about the health properties of the healthy Mediterranean diet has been the driving force for the current regeneration of users’ and market’s awareness for WEPs [56]. 

In another sense, the number of participants is relatively low for such a long period of data collection and the duration of the study is very long, which is a limitation of the study. Nevertheless, the ethnobotanical indices employed and the developed methodology enable the comparison of data across various areas and studies, thereby offering novel insights into plants used in gastronomy.

## 4. Materials and Methods

### 4.1. The Study Area

The Region of Valencia has a total surface of 23,255 km^2^ and it is situated on the east coast of Spain, between the coordinates UTM 702,000 N and 4,288,000 E. The Valencia Region has a very complex topographical feature. The region is divided into 3 provinces: Alicante, Valencia, and Castellón. The coastal area is occupied by a series of littoral plains, with the high relief to the south and southwest being made by the Betica Mountains. Their easterly area, namely, the Pre-Betica range, reaches directly into the Mediterranean sea with cliffs and ridges of more than 700 m above MSL (mean sea level) in height near the cape of San Antonio. To the north and northwest are the Spanish mountains with a high elevation (Javalambre) and extensive mesas (Gúdar), both just over 2000 m above MSL. Almost exactly west of Valencia, the elevations are lower, with the highest points reaching only 1100 m MSL, and they have direct elevation and almost no ridges from the coast to the low plateau. The climate of the region is predominantly semi-arid with annual rainfall ranging from 300 to 500 mm. The south is more arid with less than 320 mm of rainfall. There are two areas with annual rainfall above 850 mm: the south of the Valencian Gulf and the northwestern limit of the territory [57].

### 4.2. Ethnobotanical Interviews and Data Treatment

A total of 58 municipalities were prospected with oral interviews in different locations of the Valencian Community between 1999 and 2023. Spanish and Catalan vernacular names of traditionally used plant species were collected in the field through meetings with local people. Ethnobotanical information is mainly based on semi-structured interviews in which we collect information such as different plants used for culinary purposes. Moreover, the information gathered in conversations was further verified by field observations with the stakeholders. This type of research is, in sociological expressions, known as “participatory observation” [8]. A statistical analysis of the results has been carried out using Statgraphics centurion XVIII software.

Local knowledge for edible species used in gastronomy was collected from 203 women and 139 men (342 records) (Figure 3). People with a specific profile were carefully chosen in order to acquire high quality and consistent data. The age of the volunteers was in the range of 26–95 years, with more than 65% being over 60 years of age. Ethnobotanical informants live in a rural location and from a range of socioeconomical strata, who had used edible species throughout their lives.

On the other hand, ethical principles of the International Society of Ethnobiology were taken into account [40]. Thus, this study was carried out considering these anonymous surveys, carried out with a survey that does not contain the identity of the respondent and did not affect the privacy of the person. Participation in this questionnaire was voluntary, after an oral description by the researcher about the content of the questionnaire and the objectives of the survey. Participants signed a consent form agreeing to participate in this study.

A digital voice recorder was used to record ethnobotanical conversations and generate an audio database in mp3 format of the ethnobotanical material with a total of 211 h of information obtained. Moreover, a photo archive, with photographs of each species mentioned by the participant, has been compiled and kept in the Earth and Environmental Sciences Department Collection of Alicante University. Plant species were collected from different parts of the Valencian Community and were taxonomically identified in the laboratory using a detailed local dichotomous key and floras [58] and registered into ABH (official Herbarium of Alicante University). In addition, we used Excel^®^ 2019 to perform a simple statistical analysis of the collected data. Additionally, Fisher’s LSD method has been used in the ANOVA to create confidence intervals for all pairwise differences between factor level means, while controlling the individual error rate at a specified level [52]. As a result, we calculated the relative frequency of citation (RFC) at which each species of plant was used for its culinary purposes (Equation (1)). Furthermore, we obtained a cultural importance index (CI) where each edible plant is given a value according to its relative importance (Equation (2)). Furthermore, we included cultural value index (CV) that measures the cultural and practical values to capture different features of the importance of a plant species for persons (Equation (3)) [44,59]. Using the Jaccard similarity coefficient (Cj), we compared the differences between the three provinces of the Valencian Community (Alicante, Valencia, and Castellón) [42].

Relative frequency of citation (*RFC*). This index reflects the local relative importance of each species. It is calculated by dividing the citation frequency (*FC*) by the population size (*N*) [59].
(1)$$RFC_{s}=\frac{FC_{s}}{N}=\frac{\sum_{i=i_{1}}^{i_{N}}UR_{i}}{N}$$

Equation (1). RFC formula.

Cultural importance index (*CI*). This index takes into account not only the spread of the use (number of informants) for each species, but also its versatility [59].
(2)CIS=∑u=u1uNC∑i=i1iNURuiN

Equation (2). *CI* formula.

Cultural value index (*CV*). The index of cultural value captures the theoretical importance of a plant for a given culture [60].
(3)CVS=NUSNC × FCSN × ∑u=u1uNC∑i=i1iNURuiN

Equation (3). *CV* formula.

The Jaccard coefficient of similarity (*Cj*) has been used to compare internal differences [50].
(4)$$Cj=\frac{c}{a+b+c}$$

Equation (4). *Cj* formula.

## 5. Conclusions

In conclusion, data on the plant species commonly used as food in the Valencian Community were little recognized and studied. The initial hypothesis is fulfilled because there is a traditional knowledge about the use of edible plants in Mediterranean environments and it is confirmed that they can be used in today’s cuisine. Thus, authors collected traditional recipes utilizing local plants, coming from the local heritage and showing many specificities of plant species related to gastronomy, facilitating access to remarkable and novel data. Moreover, women continue to play a key role in the preservation of traditional culinary patrimony in the present. Furthermore, some wild plants such as *Asparragus acutifolius*, *Cichorium intybus*, *Foeniculum vulgare*, *Eryngium campestre*, *Crataegus monogyna*, *Lavatera arborea*, *Papaver rhoeas* or *Allium roseum* could be cultivated without too many agronomic requirements and be useful for the local economy. It is, therefore, a phytogenetic resource that must be conserved and valued so that new generations can enjoy it in the future. Finally, further studies are needed to complete the knowledge on ethnobotanical exploitation in the study area and to raise awareness of this natural resource.

## Figures and Tables

**Figure 1 plants-13-00775-f001:**
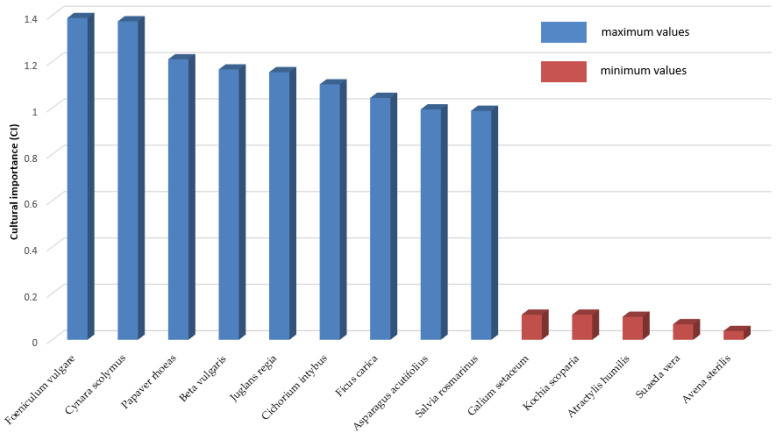
Species of major and minor cultural importance (CI).

**Figure 2 plants-13-00775-f002:**
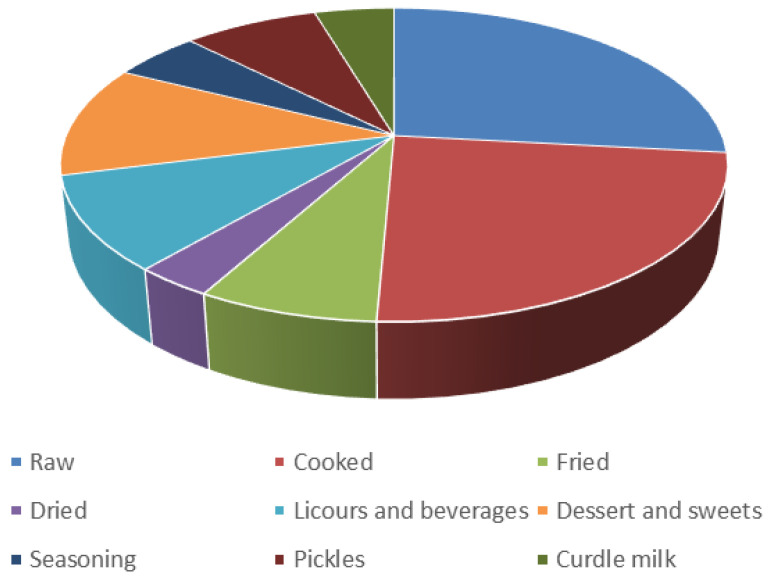
Different ways of using edible plants for culinary purposes.

**Figure 3 plants-13-00775-f003:**
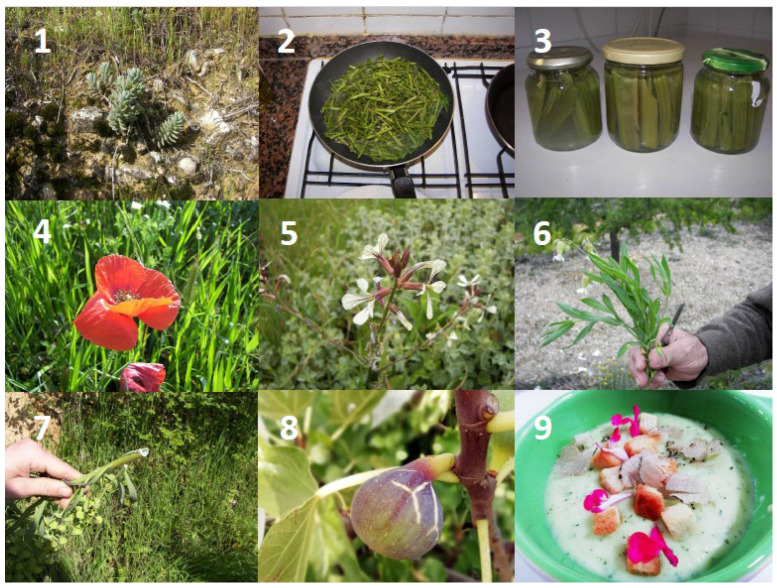
Images of representative plants in the study area. (**1**) *Sedum sediforme*, (**2**) *Asparagus acutifolius*, (**3**) *Cynara cardunculus*, (**4**) *Papaver rhoeas*, (**5**) *Eruca vesicaria*, (**6**) *Silene vulgaris*, (**7**) *Euphorbia characias*, (**8**) *Ficus carica,* and (**9**) *Salvia microphylla*.

**Figure 4 plants-13-00775-f004:**
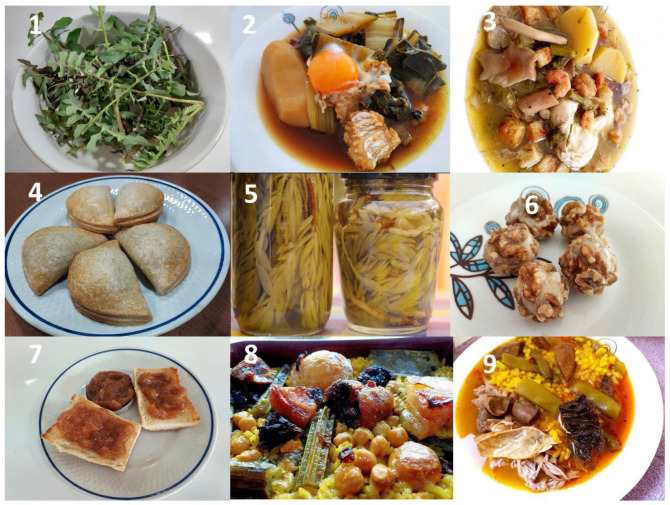
Images of traditional recipes with wild and cultivated plants with their RFC. (**1**) *Sonchus tenerrimus* salad, (**2**) “Borreta” is *Beta vulgaris* soup, (**3**) asparagus with mushroom soup, (**4**) vegetable pastries, (**5**) *Sedum sediforme* pickles, (**6**) sweet walnuts, (**7**) fig jam, (**8**) baked rice with cardoons, (**9**) rice broth with *Silene vulgaris*, meat, and mushrooms.

**Figure 5 plants-13-00775-f005:**
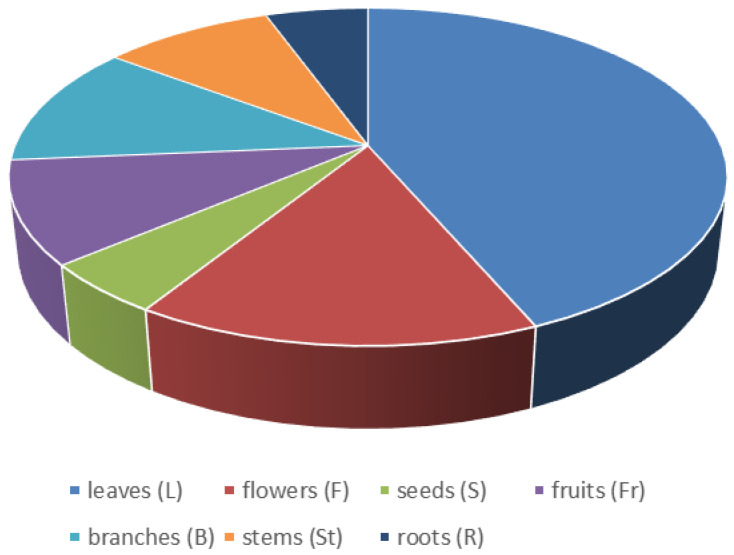
Plant part used (no. of species).

**Figure 6 plants-13-00775-f006:**
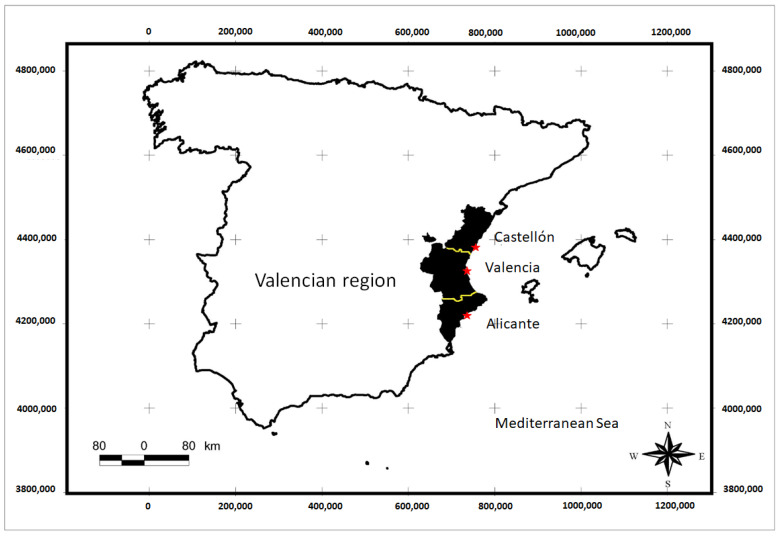
Study area map (Valencia Region, Spain).

**Figure 7 plants-13-00775-f007:**
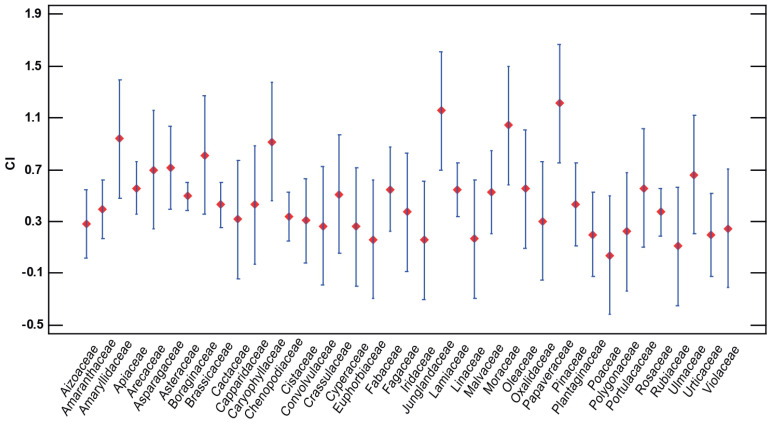
LSD 95% (least significant difference) Fisher means plot for CI vs. family.

**Figure 8 plants-13-00775-f008:**
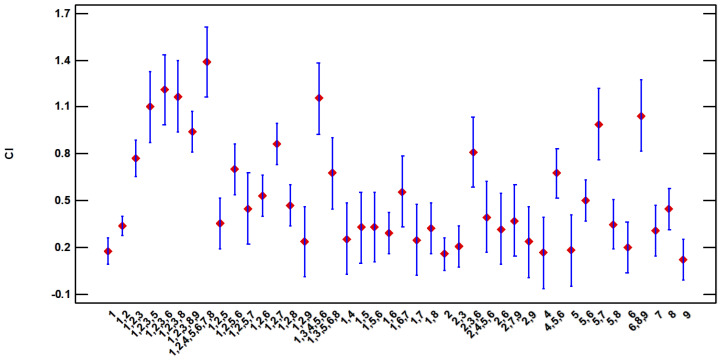
LSD 95% Fisher means plot for CI vs. gastronomic uses.

**Figure 9 plants-13-00775-f009:**
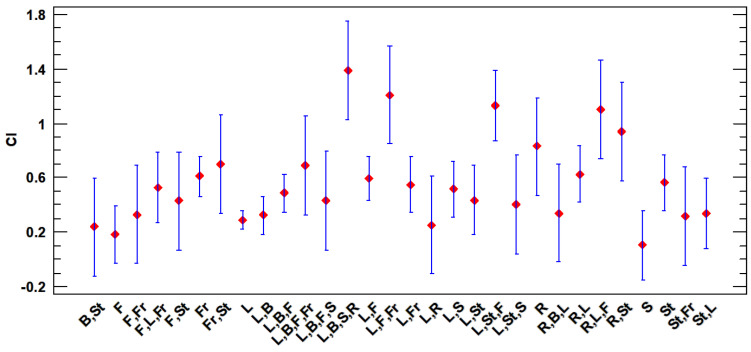
LSD 95% Fisher means plot for CI vs. plant part used.

**Figure 10 plants-13-00775-f010:**
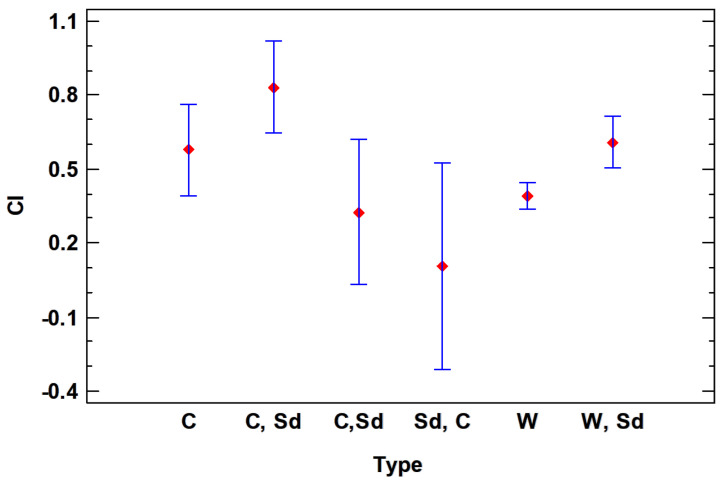
LSD 95% Fisher means plot for CI vs. type.

**Table 1 plants-13-00775-t001:** Edible plant species in the Valencian Community and their traditional culinary uses.

Family	Scientific Name	ABH	Spanish Name	Catalan Name	Uses	Part Used	RFC	CI	CV	Type
Aizoaceae	*Aizoon hispanicum* L.	8129	Aguazul	Gasul	1,2	L	0.11	0.137	0.003	W
	*Carpobrotus edulis* (L.) N.E. Br	1663	Uña de gato, hierba del cuchillo	Platanera de mar, bàlsam	1,2,6	L, Fr	0.3	0.415	0.042	W
	*Mesembryanthemum crystallinum* L.	30,668	Gazul	Aiguasul, aiguasula	1	L	0.28	0.281	0.009	W, Sd
Amaranthaceae	*Amaranthus retroflexus* L.	47,631	Moco-pavo	blets	1,2	L, S	0.36	0.409	0.033	W
	*Arthrocnemum macrostachyum* (Moric.) C.Koch	1311	Salicornia, sosa	Sosa grossa, cirialera	1,2,8	L	0.29	0.360	0.034	W
	*Salicornia ramosissima* J. Woods	21,146	Hierba salada	Cirialera	1,8	L, B	0.22	0.325	0.016	W
	*Salsola soda* L.	40,583	Barrilla común, álcali	Sosa	1,2,8	L, B	0.3	0.482	0.049	W, Sd
	*Allium roseum* L.	43,307	Ajo de culebra	All bord	1,2,7	R, St	0.49	0.936	0.154	W
Apiaceae	*Crithmum maritimum* L.	12,293	Cresta marina	Fenoll marí	8	St, L	0.37	0.368	0.015	W
	*Daucus carota* L.	33,104	Zanahoria silvestre	Pastanaga	1,2	R, L	0.2	0.237	0.010	W
	*Eryngium campestre* L.	11,155	Cardo corredor	Panical	1,2,5	R, B, L	0.3	0.339	0.033	W
	*Foeniculum vulgare* Mill.	23,129	Hinojo	Fenoll	1,2,4,5,6,7,8	L, B, S, R	0.83	1.389	0.900	W, Sd
	*Petroselinum crispum* (Mill.) A. W. Hill	31,693	Perejil	Jolivert	7	L, S	0.46	0.456	0.023	C, Sd
Arecaceae	*Chamaerops humilis* L.	539	Palmito	Margalló	1,2,6	Fr, St	0.49	0.699	0.114	W, Sd
Asparagaceae	*Asparagus acutifolius* L.	37,509	Espárrago	Esparreguera	1,2,3	St	0.87	0.994	0.287	W
	*Asparagus horridus* L.	31,836	Espárrago	Esparreguera	1,2,3	St	0.46	0.433	0.067	W
Asteraceae	*Atractylis humilis* L.	19,668	Cardo heredero	-	9	L	0.1	0.099	0.001	W
	*Carthamus lanatus* L.	18,999	Cardo chico, azotacristos	Card sant	2	L	0.17	0.170	0.003	W, Sd
	*Centaurea calcitrapa* L.	17,847	Cardo estrellado	Obriülls	1,2,9	B, St	0.19	0.237	0.015	W
	*Chondrilla juncea* L.	19,736	Chicoria	Morrets	1,2	L, St	0.25	0.316	0.018	W
	*Cichorium intybus* L.	37,547	Achicoria silvestre	Cama-roja	1,2,3,5	R, L, F	0.87	1.102	0.428	W
	*Cynara cardunculus* L.	35,991	Cardo de comer	Penca	1,2,3,8,9	L, St, F	0.84	0.883	0.412	C, Sd
	*Cynara scolymus* L.	31,715	Alcachofera	Carxofera	1,2,3,8,9	L, St, F	0.96	1.374	0.732	C, Sd
	*Helminthotheca echioides* (L.) Holub	40,232	Raspayo, cardo perruno	Arpell, coleta, llengua de bou	1,2	L	0.25	0.278	0.015	W
	*Lactuca serriola* L.	47,376	Lechuga silvestre	Encisam bord, lletugueta	1	L	0.23	0.225	0.006	W
	*Limbarda crithmoides* (L.) Dumort.	8205	Hierba del cólico	Salviò	1,8	L	0.25	0.316	0.018	W
	*Reichardia intermedia* (Sch. Bip.) Samp.	16,443	-	Cosconella, herba dolça	1,2	L	0.17	0.173	0.007	W
	*Scolymus hispanicus* L.	20,754	Cardo	Card	2,7,9	L, F, B	0.34	0.371	0.042	W
	*Scolymus maculatus* L.	20,114	Cardo, cardillo	Card	2,9	L, B	0.23	0.234	0.012	W
	*Scorzonera hispanica* L.	4557	Escorzonera	Escorçonera	1,2,6	R, L	0.25	0.474	0.040	W
	*Silybum marianum* (L.) Gaertn.	32,020	Cardo mariano	Card marià	1,2,3,8,9	L, B, F	0.52	0.561	0.161	W
	*Sonchus oleraceus* L.	47,365	Cerrajas, llicsones	Lletsó d’ase	1,2	L	0.55	0.608	0.074	W
	*Sonchus tenerrimus* L.	37,483	Cerrajas, llicsones	Lletsó de pardalet	1,2	L	0.65	0.757	0.109	W
	*Taraxacum vulgare* (Lam.) Schrank	1808	Diente de león	Dent de lleó	1,2,3	L	0.59	0.746	0.146	W
Boraginaceae	*Borago officinalis* L.	6950	Borraja	Borratja	2,3,6	L, F	0.65	0.810	0.174	W, Sd
Brassicaceae	*Cakile maritima* Scop.	41,661	Oruga	Rave marí	1,4	L, R	0.25	0.254	0.014	W
	*Capsella bursa-pastoris* (L.) Medicus	47,380	Zurrón de pastor	Borsa de pastor	1,2	L	0.2	0.231	0.010	W
	*Diplotaxis erucoides* (L.) DC.	47,963	Rabaniza	Citró	1,2,7	L, F	0.56	0.749	0.141	W
	*Eruca vesicaria* (L.) Cav.	41,713	Rúcula	Ruca	1,2,7	L, F	0.66	0.909	0.200	W, Sd
	*Lobularia maritima* (L.) Desv.	1522	Mastuerzo	Caps blancs	1,7	L, F	0.23	0.249	0.012	W
	*Moricandia arvensis* (L.) DC.	38,239	Collejón	Collextó	2	L	0.18	0.178	0.004	W
	*Rorippa nasturtium-aquaticum* (L.) Hayek subsp. *nasturtium-aquaticum*	38,575	Berros	Créixens	1,2	L, B	0.32	0.433	0.031	W
Cactaceae	*Opuntia maxima* Mill.	16,138	Chumbera	Figuera de pala	2,6	St, Fr	0.3	0.316	0.021	C, Sd
Capparidaceae	*Capparis spinosa* L.	1275	Alcaparra	Tàpenes	8	F, St	0.43	0.427	0.020	W
Caryophyllaceae	*Silene vulgaris* (Moench) Garcke subsp. *vulgaris*	20,083	Collejas	Conillets	1,2,3	L	0.68	0.915	0.208	W
Chenopodiaceae	*Atriplex halimus* L.	3386	Salado blanco, orgaza	Salat blanc	1	L	0.13	0.135	0.002	W, Sd
	*Beta vulgaris* L.	10,652	Acelga	Bleda	1,2,3,8	R, L	0.91	1.167	0.472	W, Sd
	*Chenopodium murale* L.	7623	Cenizo, salao	Blet de paret	1,2	L, St, S	0.33	0.401	0.029	W
	*Halimione portulacoides* (L.) Aellen.	32,375	Cenizo blanco, sabonera	Verdolaga marina	1,2	L	0.13	0.152	0.004	W
	*Kochia scoparia* (L.) Schrad.	32,400	Mirabel, ciprés de verano	Bellverd, mirambell	1	L	0.11	0.108	0.001	Sd, C
	*Suaeda vera* Forssk. ex J. F. Gmel.	17,190	Sosa	Sosa, salat	1	L	0.07	0.067	0.001	W
Cistaceae	*Cistus albidus* L.	43,248	Jara blanca	Estepa	5,6	L, B, F, S	0.33	0.430	0.032	W
	*Cistus clusii* Dunal	33,909	Romero macho	Matagall	5	L, B, F	0.18	0.181	0.004	W
Convolvulaceae	*Convolvulus arvensis* L.	5857	Correhuela, campanilla	Corriola, campaneta	6	F	0.27	0.266	0.008	W
Crassulaceae	*Sedum sediforme* (Jacq.) Pau subsp. *sediforme*	15,304	Uva de pastor	Raïm de pastor	5,8	L, B	0.46	0.509	0.053	W
Cyperaceae	*Scirpus holoschoenus* L. subsp. *holoschoenus*	23,224	Juncos	Juncs	1	St	0.26	0.257	0.007	W
Euphorbiaceae	*Euphorbia characias* L.	19,690	Lecheterna	Lletera	9	L, B	0.16	0.161	0.003	W
Fabaceae	*Glycyrrhiza glabra* L.	11,289	Regaliz	Regalissia	4,5,6	R	0.77	0.830	0.213	W, Sd
	*Scorpiurus muricatus* L.	867	Alacranera	Llengua d’ovella	1,2	L	0.27	0.266	0.016	W
Fagaceae	*Quercus ilex* L. subsp. *rotundifolia* (Lam) Schwartz ex T. Morais	20,160	Carrasca	Carrasca	1,2,5	Fr	0.27	0.371	0.034	W, Sd
Iridaceae	*Crocus serotinus* subsp. *salzmannii* (J. Gay) B. Mathew	44,757	Azafrán silvestre	Safrà	7	F	0.15	0.155	0.003	W
Junglandaceae	*Juglans regia* L.	32,699	Nogal	Noguera	1,3,4,5,6	Fr	0.77	1.155	0.493	C
Lamiaceae	*Mentha spicata* L.	22,151	Hierbabuena	Herba-sana	1,6,7	L, B, F	0.45	0.558	0.084	C
	*Ocimum basilicum* L.	23,959	Albahaca	Alfàbega	1,2,5,7	L	0.33	0.450	0.067	C
	*Salvia lavandulifolia* subsp. *mariolensis* (Figuerola) Alcaraz & De la Torre	4663	Salvia de Mariola	Sàlvia de Mariola	1,5	L, B, F	0.29	0.327	0.021	W
	*Salvia microphylla* Kunth	43,610	Hierba de mirto	Sogra i nora	5,6	L, B, F	0.34	0.404	0.031	C, Sd
	*Salvia rosmarinus* Schleid.	37512	Romero común	Romer	5,7	L, B, F	0.63	0.988	0.138	W, Sd
Linaceae	*Linum narbonense* L.	22671	Lino	Llinós	4	S	0.17	0.167	0.003	C
Malvaceae	*Lavatera arborea* L.	8334	Malva arbórea	Malva vera	1,6	F, L, Fr	0.3	0.336	0.023	C, Sd
	*Malva sylvestris* L.	37,062	Malva común	Malva	1,2,5,6	F, L, Fr	0.6	0.716	0.192	W
Moraceae	*Ficus carica* L.	47,519	Higuera común	Figuera	6,8,9	L, Fr	0.91	1.044	0.315	C, Sd
Oleaceae	*Olea europaea* L.	17,212	Olivo	Olivera	8	Fr	0.55	0.553	0.034	C
Oxalidaceae	*Oxalis pes-caprae* L.	38,242	Agrios, agritos	Agrets	1,6	St, L	0.26	0.301	0.017	W
Papaveraceae	*Papaver rhoeas* L.	37,589	Amapola	Rosella	1,2,3,6	L, F, Fr	0.6	1.211	0.322	W
Pinaceae	*Pinus halepensis* Mill.	37,506	Pino carrasco	Pi blanc	5,8	L, Fr	0.14	0.190	0.006	W, Sd
	*Pinus pinea* L.	32,768	Pino piñonero	Pi pinyoner	1,3,5,6,8	L, S	0.55	0.675	0.207	W, Sd
Plantaginaceae	*Plantago albicans* L.	34,356	Llantén	Herba-fam	2	L	0.18	0.181	0.004	W
	*Plantago coronopus* L.	21,822	Cervina, estrellamar	Cervina	2	L	0.22	0.219	0.005	W
Poaceae	*Avena sterilis* L.	810	Avena loca	Cogula	2	S	0.04	0.038	0.000	W
Polygonaceae	*Rumex pulcher* L.	15,055	Romanza	Cama-roges	2,3	L	0.2	0.222	0.010	W
Portulacaceae	*Portulaca oleracea* L.	36,619	Verdolaga	Verdolaga	1,2,8	L, St	0.44	0.556	0.081	W
Rosaceae	*Crataegus monogyna* Jacq.	43,322	Espino blanco	Espinal blanc	4,5,6	Fr	0.45	0.520	0.079	W
	*Fragaria vesca* L.	52,157	Fresa	Maduixa	2,4,5,6	Fr	0.33	0.395	0.057	W, Sd
	*Rosa agrestis* Savi	51,473	Rosal silvestre	Roser bord, gavarrera	1,5,6	F, Fr	0.21	0.330	0.024	W
	*Rubus ulmifolius* Schott	40,230	Zarzamora	Esbarzer	1,2,5,6	L, B, F, Fr	0.52	0.690	0.159	W
	*Sanguisorba minor* Scop.	42,017	Pimpinela	Pimpinel·la	1	L	0.16	0.164	0.003	W
	*Sorbus aria* (L.) Crantz	3530	Mostazo	Serval, Moixera	6	F	0.13	0.132	0.002	W
Rubiaceae	*Galium setaceum* Lam.	36,674	Jabonera	Sabonera	9	L, B	0.11	0.108	0.001	W
Ulmaceae	*Celtis australis* L.	52,135	Almez	Llidoner	5,6	Fr	0.57	0.658	0.083	W
Urticaceae	*Urtica dioica* L.	40,147	Ortiga	Guardians, picapatos	2,3	L	0.16	0.205	0.007	W
	*Urtica urens* L.	33,640	Ortiga menor	Guardians, picapatos	2,3	L	0.14	0.184	0.006	W
Violaceae	*Viola odorata* L.	43,262	Violeta	Viola	1,6	L, F	0.19	0.246	0.011	W, Sd

Scientific names and families. Herbarium Voucher code (ABH). Spanish and Valencian vernacular names. Gastronomic uses: raw (including salads) (1), cooked (2), fried (3), dried (4), liquors and beverages (5), dessert and sweets (6), seasoning (7), pickles (8), and curdle milk (9). Plant part used: leaves (L), flowers (F), seeds (S), fruits (Fr), branches (B), stems (st), and roots (R). RFC: relative frequency of citation. CI: cultural importance index. CV: cultural value index. Type: W, wild; C, cultivated or Sd, semi-domesticated.

## Data Availability

The species mentioned in the ethnobotanical analysis can be consulted virtually in the repository of the University of Alicante at “https://herbariovirtual.ua.es/ (accessed on 16 December 2023)”.

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
