# Peer review of "Long-Term Monitoring of the Traditional Knowledge of Plant Species Used for Culinary Purposes in the Valencia Region, South-Eastern Spain"

_plants, 2024, doi:10.3390/plants13060775_

Round 1

Reviewer 1 Report (Previous Reviewer 2)

Comments and Suggestions for Authors

lines 37-39 Please, add references

line 46 Does it mean that due to food scarcity there was overharvesting of wild plants which led to their diminishing in the nature, subsequently decline in there use in recent days?

line 48 please give examples for recipes which are popular

line 51 please use a synonym to “occasion”

line 54-56 This sentence still has grammar mistakes ("wild-foraged foods very healthy") and it is difficult to comprehend.

line 56-58 “These natural products…”  did you mean the plants? They are not products, besides I believe it will be more beneficial if you use examples for plant species or families with specific bio active substances, with proven beneficial effect on human and/or animal nutrition/health.

line 59-63 Again, it will be more beneficial if you give examples, not just generalization, because not all the readers are familiar with the Mediterranean flora in Europe

line 64  please, re-phrase “procedure of oral transmission has broken down”

line 67-68 This sentence is difficult to comprehend. Please, re-phrase it

The following sentences also need to be polished, in order to explain better the relationship between the different research approaches and the interdisciplinary nature of the topic.

The hypothesis needs to be more focused, because it is well known that plants are traditionally well presented in the Mediterranean cuisine 

line 76 please replace “remarkable” with more suitable word

In the result section I will suggest to start the participants and then the findings with the tables, figures, graphics etc.

In Fig 4, is better if you follow the same color of the indexes like in Fig. 3

lines 89-91 the  description of participants  needs to be re-phrased, because it remains unclear what the authors exactly want to point out with “highlighting the relevance of this characteristic  in the traditional cuisine, compared to the typical role of men in rural tasks”. Also, did the authors have info about the occupation/ education level of the participants?

line 105 instead of “most characterized” is it not better to use most commonly used?

Lines 1267-129 What is the idea of the 15 species mentioned in this sentence? Can you clarify?

Lines 134 -144 Why don’t you put the group designations in brackets and write the culinary use in full? For example “Firstly, the most common gastronomic use is group 1, with 58 plant species known  to be used as raw vegetables or salads “ might become “ The most common gastronomic use of plants was raw or for salads(group 1) with 58 species.”

Lines 153-172 It is good that the authors added traditional dishes where the local plants are used, but I saw no references for the recipes that has been described. Are those recipes entirely collected from the participants or some of them came for cooking books?

Figure 6. Please specify with arrows the location and show the border of the 3 provinces

LSD 95% abbreviation (least significant difference) needs to be written in full the first time when mentioned.

Lines 227-228 rephrase it like ” collection of wild edible plants (WEPs) is a common practice among  part of the indigenous inhabitants…”

lines 232-235 give specific examples for  plants and plant phytochemicals whit health benefits from those plants found most commonly used in that study!

Line 235-236 This sentence is a bit odd, since plants are non-replaceable part of diet due the various vitally important components (like vitamins, amino acids, carbs, fibers etc.)

line 236-237 give examples and references. In the following sentences are mentioned several plants but the amount of bioactive substances are not mentioned.

line 264-266 There are such studies  in nearly every country in Europe and many countries outside of it.

Line 273 replace ”important” with “common”

Line 273 278 this sentences needs to be fixed (for example only 1 species have English name and the rest are only in Latin) and please use studies instead of investigations.

Lines 278-279 please explain it in more details

Lines 286-291 explain the similarities and differences with the other studies in more details.

Lines 292-293 where is the comparative reference/s?

Lines 296-297 please make the sentence more comprehensive

Which species have the potential to “rejuvenate the local economy?”

What program was used for Fisher's LSD  method?

The authors did not provide info about how many respondents are from each of the 3 provinces. Besides, the number of participants is relatively low for such a long period of data collection. The time span of the study is also very long. These facts must be added as limitation of the study in the discussion section.

The entire conclusion needs to be reformed.

Comments on the Quality of English Language

Please check my notes.

Author Response

lines 37-39 Please, add references. New references have been added.

line 46 Does it mean that due to food scarcity there was overharvesting of wild plants which led to their diminishing in the nature, subsequently decline in there use in recent days? We have incorporated this information into the manuscript, including the corresponding bibliographic citation.

line 48 please give examples for recipes which are popular. We have added more information on this topic.

line 51 please use a synonym to “occasion”. We have changed the term to “opportunity”.

line 54-56 This sentence still has grammar mistakes ("wild-foraged foods very healthy") and it is difficult to comprehend. We have modified this sentence.

line 56-58 “These natural products…”  did you mean the plants? They are not products, besides I believe it will be more beneficial if you use examples for plant species or families with specific bio active substances, with proven beneficial effect on human and/or animal nutrition/health. We have modified this sentence and we added information about the specific benefits of plants in the discussion section.

line 59-63 Again, it will be more beneficial if you give examples, not just generalization, because not all the readers are familiar with the Mediterranean flora in Europe. We have added this information with the corresponding reference.

line 64 please, re-phrase “procedure of oral transmission has broken down”. We have modified it.

line 67-68 This sentence is difficult to comprehend. Please, re-phrase it. We have re-phrased it.

The following sentences also need to be polished, in order to explain better the relationship between the different research approaches and the interdisciplinary nature of the topic. We have modified it.

The hypothesis needs to be more focused, because it is well known that plants are traditionally well presented in the Mediterranean cuisine. We have modified it.

line 76 please replace “remarkable” with more suitable word. We have modified it.

In the result section I will suggest to start the participants and then the findings with the tables, figures, graphics etc. We have modified it.

In Fig 4, is better if you follow the same color of the indexes like in Fig. 3. We have modified it.

lines 89-91 the description of participants needs to be re-phrased, because it remains unclear what the authors exactly want to point out with “highlighting the relevance of this characteristic in the traditional cuisine, compared to the typical role of men in rural tasks”. Also, did the authors have info about the occupation/ education level of the participants? This has been better explained and information on the occupation and educational level of the participants has been added.

line 105 instead of “most characterized” is it not better to use most commonly used? We have modified it.

Lines 127-129 What is the idea of the 15 species mentioned in this sentence? Can you clarify? This has been clarified in the manuscript, but corresponds to the plant species sold in local markets.

Lines 134 -144 Why don’t you put the group designations in brackets and write the culinary use in full? For example: “Firstly, the most common gastronomic use is group 1, with 58 plant species known to be used as raw vegetables or salads “might become “The most common gastronomic use of plants was raw or for salads (group 1) with 58 species.” This has been modified in accordance with your comments.

Lines 153-172 It is good that the authors added traditional dishes where the local plants are used, but I saw no references for the recipes that has been described. Are those recipes entirely collected from the participants or some of them came for cooking books? The recipes shown have been provided by the people interviewed, but there are different publications on traditional cuisine that include some of these plants. We have modified the text and we have included references.

Figure 6. Please specify with arrows the location and show the border of the 3 provinces. Figure 6 has been modified to include the requested information.

LSD 95% abbreviation (least significant difference) needs to be written in full the first time when mentioned. We have modified it.

Lines 227-228 rephrase it like ”collection of wild edible plants (WEPs) is a common practice among  part of the indigenous inhabitants…” We have modified it.

lines 232-235 give specific examples for plants and plant phytochemicals whit health benefits from those plants found most commonly used in that study! We have added this information based on your comments and the bibliography. In general, this section has been modified according to your enriching contributions.

Line 235-236 This sentence is a bit odd, since plants are non-replaceable part of diet due the various vitally important components (like vitamins, amino acids, carbs, fibers etc.) We have modified it. In general, this section has been modified according to your enriching contributions.

line 236-237 give examples and references. In the following sentences are mentioned several plants but the amount of bioactive substances are not mentioned. In general, this section has been modified according to your enriching contributions.

line 264-266 There are such studies in nearly every country in Europe and many countries outside of it. We have modified it and we have included new references.

Line 273 replace ”important” with “common” We have modified it.

Line 273 278 this sentences needs to be fixed (for example only 1 species have English name and the rest are only in Latin) and please use studies instead of investigations. We have modified it.

Lines 278-279 please explain it in more details. We have modified it.

Lines 286-291 explain the similarities and differences with the other studies in more details. We have included new references.

Lines 292-293 where is the comparative reference/s? We have modified it. Reference “38”.

Lines 296-297 please make the sentence more comprehensive. We have modified it.

Which species have the potential to “rejuvenate the local economy? Some potential wild plant species to be cultivated in the study territory are: Mesembryanthemum crystallinum, Salicornia ramosissima, Crithmum maritimum, Cichorium intybus, Sonchus tenerrimus, Taraxacum vulgare and Borago officinalis.

What program was used for Fisher's LSD method? A statistical analysis of the results has been carried out using Statgraphics centurion XVIII software.

The authors did not provide info about how many respondents are from each of the 3 provinces. Besides, the number of participants is relatively low for such a long period of data collection. The time span of the study is also very long. These facts must be added as limitation of the study in the discussion section. Of the people interviewed, 118 were from the province of Alicante, 110 from Valencia and 114 from Castellón. The number of participants is relatively low for such a long period of data collection and the duration of the study is also very long, which is a limitation of the study.

The entire conclusion needs to be reformed. We have modified it.

Reviewer 2 Report (New Reviewer)

Comments and Suggestions for Authors

This manuscript aimed to monitor plant species used for culinary purposes in a region of Spain, providing novel information on edible plants registered for more than two decades. The quality was improved by addressing the comments from the previous reviewer.  Some comments are as follows:

1. The name of the Y-axis in Figure 1 was missing.

2. The name of Figure 2 should be revised.

3. Where are the images in Figure 3 and Figure 4 from? The sources should be mentioned.

Author Response

  1. The name of the Y-axis in Figure 1 was missing. We have modified it.
  2. The name of Figure 2 should be revised. We have modified it.
  3. Where are the images in Figure 3 and Figure 4 from? The sources should be mentioned. The images used in figures 3 and 4 are taken by the authors during the sampling process.

Round 2

Reviewer 1 Report (Previous Reviewer 2)

Comments and Suggestions for Authors

Lines 44-46This 2 sentences needs to be reformed to better fit in the text.

Lines 55-65 This part can be reduced because there is not so much information in it. Whenever the authors mentioned medicinal plants, or highly nutritious plants or those entreated with extinction etc. it is advisable to give examples.

Line 75 It has been stopped?  What exactly does that mean?

Lines 77-84 these sentences are a bit unclear

Once again the hypothesis is too generalized. It is not such a secret that there is “traditional knowledge about the use of plants for gastronomic purposes in Mediterranean environments”  Maybe you should turn your focus toward the local community knowledge about the traditional gastronomic use of wild and cultivated plant native for the region of Valencia.

lines 92 probably use of “helpful to preserve…” is better than “ outstanding”

Line100 please replace “interrogate “with “interviewed” and rearrange the entire sentence because it is a little unclear

Lines 99-110 please remove the statements about the higher level of traditional knowledge of the female participants from this part of the text and leave it for the discussion.

line 116 benefits people?

why table 1 is not here?

line 154 Is it not better to put “that have been used” 

Line 157 add “group” after most important

Lines 266-267 the previous sentence does not imply that the indigenous people are from rural area please clarify it

Lines 270 -273 rewrite this sentence because it remains a bit unclear and somehow incorrect

lines 279-281 which bacteria?, what type of cancer? Whenever you give example give it in full so that anyone can understand properly the health benefits of the use of this plant

lines 289 -293not all the effects are contributed to all the phytochemicals. Besides, each part of the plant has more or less different chemical composition so such generalization is not appropriate.

Same thing is valid for the rest of the examples given by the authors.

lines 374 377 Why the authors suggest that this species have potential to support the local economy? Please, explain in details.

lines 393-394 According to the information in the text  the average age of the participant's  is 65.7 years, with more than 65% being over 60 years of age, so statement like this should not be used unless if you have solid proofs. Additionally, the number of participants is very low so general conclusions must be avoided.

lines 410 412 rephrase the sentence to become clearer

Conclusions needs improvement, adding the fact that authors collected traditional recipes utilizing local plants, also the idea for plant species potentially useful for the local economy needs to be expanded with specific species and their advantages which makes the suitable for that purpose

Comments on the Quality of English Language

Please, check my notes.

Author Response

Lines 44-46 This 2 sentences needs to be reformed to better fit in the text. It has been modified according to your comments.

Lines 55-65 This part can be reduced because there is not so much information in it. Whenever the authors mentioned medicinal plants, or highly nutritious plants or those entreated with extinction etc. it is advisable to give examples. We have shortened this part of the text and the following sentences show some examples of useful plants.

Line 75 It has been stopped?  What exactly does that mean? The term has been revised to "interrupted." This adjustment reflects the fact that numerous individuals with extensive ethnobotanical knowledge have passed away, and the younger generation shows limited interest in delving into the subject.

Lines 77-84 these sentences are a bit unclear. It has been modified according to your comments.

Once again the hypothesis is too generalized. It is not such a secret that there is “traditional knowledge about the use of plants for gastronomic purposes in Mediterranean environments”  Maybe you should turn your focus toward the local community knowledge about the traditional gastronomic use of wild and cultivated plant native for the region of Valencia. It has been modified according to your comments.

lines 92 probably use of “helpful to preserve…” is better than “ outstanding” It has been modified according to your comments.

Line100 please replace “interrogate “with “interviewed” and rearrange the entire sentence because it is a little unclear. It has been modified according to your comments.

Lines 99-110 please remove the statements about the higher level of traditional knowledge of the female participants from this part of the text and leave it for the discussion. It has been modified according to your comments.

line 116 benefits people? It has been modified according to your comments.

why table 1 is not here? We have put table 1 behind all the results, but if you find it more convenient when it is first mentioned, we change it.

line 154 Is it not better to put “that have been used” It has been modified according to your comments.

Line 157 add “group” after most important. We have added it.

Lines 266-267 the previous sentence does not imply that the indigenous people are from rural area please clarify it. We have added this information in the previous sentence.

Lines 270 -273 rewrite this sentence because it remains a bit unclear and somehow incorrect. ” It has been modified according to your comments.

lines 279-281 which bacteria?, what type of cancer? Whenever you give example give it in full so that anyone can understand properly the health benefits of the use of this plant. It has been modified according to your comments.

lines 289 -293not all the effects are contributed to all the phytochemicals. Besides, each part of the plant has more or less different chemical composition so such generalization is not appropriate. It has been modified according to your comments.

lines 374 377 Why the authors suggest that this species have potential to support the local economy? Please, explain in details. We have added this information.

lines 393-394 According to the information in the text the average age of the participant's  is 65.7 years, with more than 65% being over 60 years of age, so statement like this should not be used unless if you have solid proofs. Additionally, the number of participants is very low so general conclusions must be avoided. That sentence with general and unsubstantiated comments has been removed from the manuscript.

lines 410 412 rephrase the sentence to become clearer. It has been modified according to your comments.

Conclusions needs improvement, adding the fact that authors collected traditional recipes utilizing local plants, also the idea for plant species potentially useful for the local economy needs to be expanded with specific species and their advantages which makes the suitable for that purpose. It has been modified according to your comments.

Round 3

Reviewer 1 Report (Previous Reviewer 2)

Comments and Suggestions for Authors

line 204 add “n”  to plats

line 222 what p value is considered significant here?

line 281 the article is about testing in vitro on breast cancer cell line, so it is better to rephrase like “..anticancer therapies  against certain types of cancer for examples against the breast cancer …”

line 293 replace “This plant contains” with “There are”

lines 298 -300  probably is better to use this instead “ .. and it has anticancer activity proven against colon adenocarcinoma, breast adenocarcinoma, glioblastoma, astrocytoma and melanoma cell lines”

Author Response

line 204 add “n”  to plats It has been modified according to your comments.

line 222 what p value is considered significant here? with a value below the significance level of 0.05

line 281 the article is about testing in vitro on breast cancer cell line, so it is better to rephrase like “..anticancer therapies  against certain types of cancer for examples against the breast cancer …” It has been modified according to your comments.

line 293 replace “This plant contains” with “There are” It has been modified according to your comments.

lines 298 -300  probably is better to use this instead “ .. and it has anticancer activity proven against colon adenocarcinoma, breast adenocarcinoma, glioblastoma, astrocytoma and melanoma cell lines”It has been modified according to your comments.

This manuscript is a resubmission of an earlier submission. The following is a list of the peer review reports and author responses from that submission.

Round 1

Reviewer 1 Report

Comments and Suggestions for Authors

The paper is only a survey of a resource, and its innovation is not prominent enough.

Comments on the Quality of English Language

The paper is only a survey of a resource, and its innovation is not prominent enough.

Reviewer 2 Report

Comments and Suggestions for Authors

A lot of sentences need to be rearranged or transformed, in order to become clearer what the authors want to state.

Lines 18-21  re-arrange the sentence

Line 21 replace “raised” with more appropriate word

Line 25-26 re-arrange the sentence

Line 30 replace “interrelated” with more appropriate word

Line 32-35;36-39 re-arrange this sentences, using proper wording

Line 41-43; 44-46; 47-48 Re-write the sentences

There are more sentences that needs to be fixed so I believe that extensive English editing is in order

Also in the introduction, again is emphasized the importance of “wild” plants, while in the results there are a lot of species which also have cultured varieties. Please address that fact!

 There are no examples of traditional dishes made by local plants, as well as insufficient data on what are the compounds of the plant origin which may have positive effect on human health.

Results

Which are the plants included in every gastronomic group? It is better to begin the results with the table, so that everyone can see which plant belongs to which group.

Why the Latin names in fig 3 are in brackets?

Line 165 very close? In statistics it is either significant or non-significant

Maybe it will be better to present the results in a different way because it looks a little bit redundant in this way (see fig 6,7,8).

Please give more examples for traditional dishes where the studied plants are used, as well as more examples for bioactive compounds derived from them and used for improvement of human health.

Line 206-208 Re-phrase that sentence to become clearer

Line 245 Crete is not a country

Comments on the Quality of English Language

Please check my notes.

Reviewer 3 Report

Comments and Suggestions for Authors

Authors have revised the article and address all the comments and suggestion. The paper is now acceptable for publication in its current form